# Constrained Source-based Salience: Network-based Visualization of Deep Learning Neuroimaging Models

Ishaan Batta *, Anees Abrol, and Vince D. Calhoun

*Abstract*—Developing frameworks using high-dimensional magnetic resonance imaging (MRI) data to characterize underlying brain changes in neurological disorders is crucial and challenging. While deep learning models offer a better prediction, tracking automated higher-order explanations at the level of brain networks is harder in learned models. We introduce a novel constrained source-based salience (cSBS) framework to automatically learn and visualize multiple independently salient brain networks associated with clinical diagnostic assessments. This is achieved by performing active subspace learning (ASL) and spatially constrained independent component analysis (scICA) in the saliency space of trained convolutional neural networks (CNNs), such that the resultant components are interpretable in terms of brain network components from existing templates. By employing a robust analysis across repeated training scenarios for an Alzheimer's disease (AD) classification task, we visualize cSBS components via full-brain back-reconstruction. We show that the cSBS components and their corresponding loadings are consistent and relevant in terms of AD-related brain areas. Our approach is able to synthesize multiple objectives of utilization of high-dimensional MRI data for deep learning along with automated detection of low-dimensional representations of the consistently involved features in terms of intrinsically salient brain networks. Our framework of automated identification of consistent underlying brain subsystems associated with clinically observed assessments is an important step toward biomarker development for various clinically observed characteristics and disorders.

*Index Terms*—Neuroimaging, structural MRI, Deep Learning, Convolutional Neural Networks (CNNs), Saliency, Interpretability, Independent Component Analysis, Spatially Constrained ICA, Subspace Learning, Brain Networks, Neurological Disorders, Alzheimer's disease

## I. INTRODUCTION

HIGH-dimensional neuroimaging modalities like magnetic resonance imaging (MRI) offer a segway to analyze measures from the whole brain but also often require complex frameworks for successful analysis and interpretation. This is even more important given the importance of neuroimaging in studying brain disorders from the perspective of diagnostic predictions and identification of affected brain subsystems. In such cases, enhanced visualization is crucial to understand the manner in which they affect multiple systems in the brain. Many data-driven approaches have been developed for using MRI datasets to learn the changes involved in structural brain disorders and summarize them into biologically meaningful representations [1], [2].

Often, the MRI data patterns are summarized in the form of lower-dimensional features [3] to be used for predictive analysis in supervised standard machine learning (SML) and statistical models [2]. This is achieved by either averaging the voxel-level features from known brain areas using pre-mapped brain atlases [4], [5], feature selection methods typically used for SML models [6], or unsupervised decomposition methods that reduce the feature dimensions in a data-driven manner [3], [7]. Decomposition methods based on principal component analysis (PCA) and independent component analysis (ICA) have been developed [8] for a data-driven computation of lower dimensional meaningful brain components to be used for further disorder-related analysis. ICA-based approaches work by optimizing a higher-order statistic of mutual independence between inherent sources in the MRI signal, which are separated to get brain components [8]. Data-driven approaches have also been extended so that the source separation is guided by prior multi-dataset brain templates [9], [10] using spatial constraints during the decomposition process [9], [11] to yield biologically meaningful features for subsequent analysis or posthoc interpretation. [12]

Various approaches have been developed to take the diagnostic information into account while performing decomposition to obtain salient brain components that characterize the involved brain changes [13]–[15]. Such semi-supervised approaches perform fusion of the decomposition step with the feature importance analysis into a single computational framework to directly obtain the active brain subspaces from the saliency space of the trained model for the given disorder [13], [16], [17]. These active subspaces are known to characterize directions of the most vital collective structural changes in the brain associated with the given diagnostic variable at hand, thus enabling the identification of key brain structures or sub-systems that drive the diagnostic discrimination between controls and subjects with a particular diagnosis.

The onset of deep learning (DL) models has opened up the possibility of utilizing full-brain features along with improved prediction outcomes without having to go through any feature reduction or prior decomposition steps, yet robustly encoding the predictive information from the data in line with pathophysiological changes [18]. However, when performing a posthoc feature analysis after predictive training on MRI data,

The research presented in this work was funded by the National Institutes of Health (NIH) under grant number NIH-R01AG073949.

All authors are with the Center for Translational Research in Neuroimaging and Data Science (TReNDS): Georgia State University, Georgia Institute of Technology, and Emory University, Atlanta, USA

* Corresponding author. Email: ibatta@gatech.edu

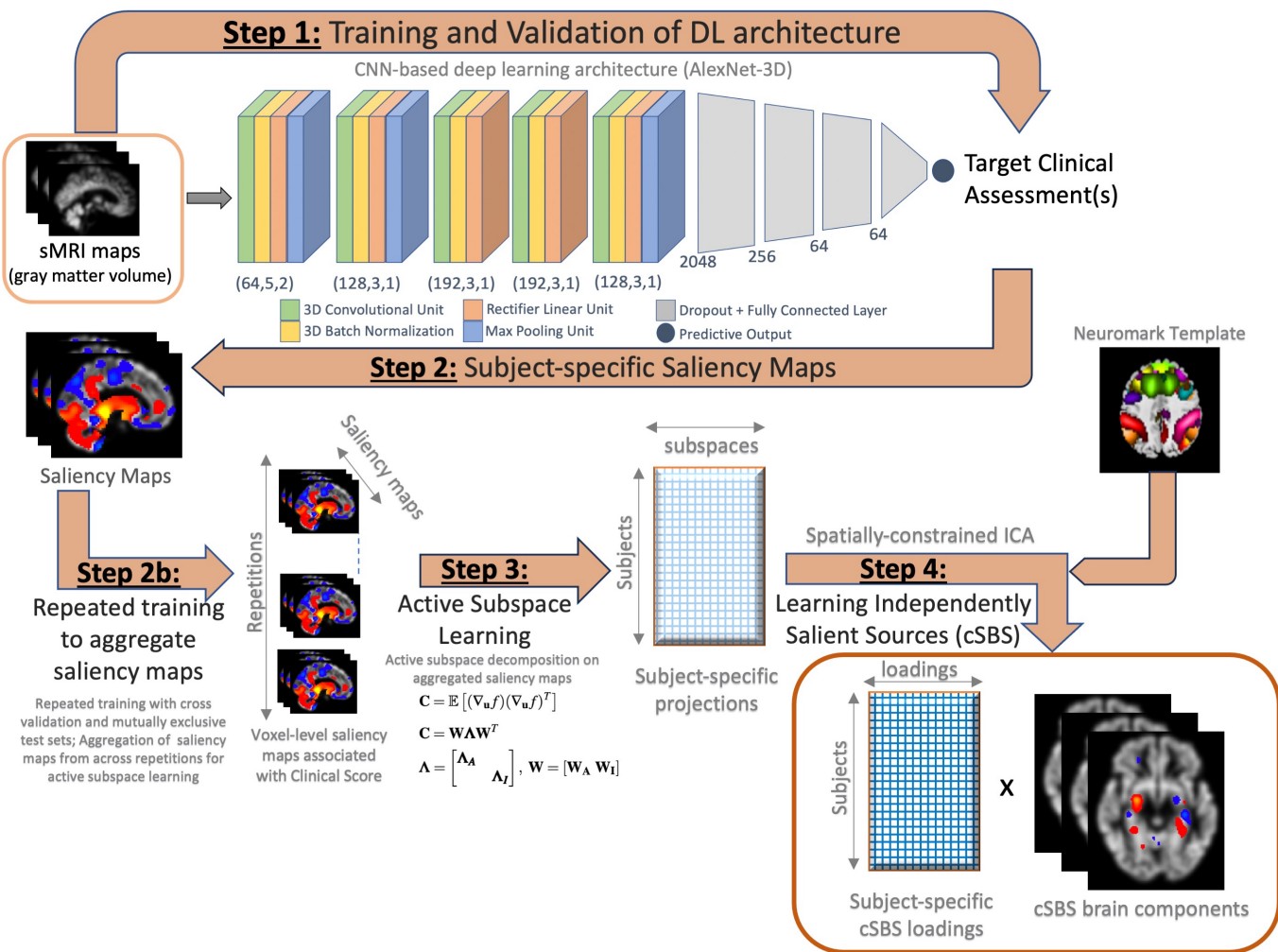

Fig. 1: Overview of the methodological pipeline for constrained source-based salience (cSBS) analysis. First, a convolutional neural network (CNN)-based DL architecture is trained as a diagnostic classifier with full-brain gray matter volume (GMV) maps as input. In the second step, subject-specific saliency maps are computed using back-propagation followed by repeated analysis on mutually exclusive test sets. These maps are then aggregated from across repetitions covering the whole dataset and used for a semi-supervised active subspace computation in the third step involving an eigendecomposition in the saliency space to select the top salient directions that drive the classification. Lastly, a spatially constrained ICA (scICA) step is performed to obtain meaningful salient brain components (called cSBS components) along with corresponding loadings signifying their contributive strength in defining diagnosis-related structural changes in gray matter volume in the brain.

DL models are either hard to interpret in terms of decoding salient features at the level of brain areas or yield subject-specific full-brain importance maps that need further manual analysis [19], leaving intact the problem of reducing data dimensions into meaningful brain components.

Our work presents a methodological framework constrained source-based salience (cSBS) analysis for structural neuroimaging data to synthesize the three-fold objective of (a) effective utilization of full-brain voxel-level structural features using deep learning, (b) semi-supervised decomposition taking target diagnostic information into account when performing source separation into structural components, and (c) performing decomposition using spatial constraints to ensure interpretability in terms of existing reference templates for brain

components. We achieve this by first training a convolutional neural network (CNN)-based DL architecture for diagnostic classification using full-brain gray matter volume (GMV) maps from an Alzheimer's disease (AD) dataset, followed by computation of subject-specific salient representation maps. These maps are aggregated from across subjects and used for a semi-supervised active subspace learning and spatially constrained ICA (scICA) step to compute meaningful salient brain components (termed as cSBS components) and corresponding loadings representing their contributive strength towards defining AD-related structural changes in GMV. Through a robust repeated posthoc analysis on an AD dataset, we show the cSBS framework successfully identifies different sources from the MRI signal with discriminatory saliency toward AD diagno-

sis that are biologically relevant in terms of interpretability through reference templates as well as comprising of brain areas and sub-domains with known involvement in AD.

## II. METHODS

### A. Dataset and Preprocessing

The dataset used for this study included ADNI (adni.loni.usc.edu) which is an Alzheimer's disease (AD) dataset. Structural MRI (sMRI) data from the first-visit baseline scans was used for performing a diagnostic classification task between Alzheimer's disease (AD) and control (CN) subjects. The first-visit data had a sample size of $800$ (aged 55-91 yrs, M/F = 404/396) with 468 CN and 332 AD subjects.

Preprocessing of the sMRI data was done using the standard preprocessing pipeline in SPM12 software as in prior studies [20]. For using as input features for the deep learning architecture, the structural gray matter volume (GMV) maps were warped into the standard MNI space with dimensions 121 x 145 x 121 and a voxel size of 1.5mm x 1.5mm x 1.5mm. This was followed by a Gaussian smoothing with FWHM = 12mm.

### B. Deep CNN Classifier and Model Training

Deep learning architectures have been shown to successfully outperform standard machine learning methods during classification tasks using high dimensional voxel-level neuroimaging data [18], [21]. The 3-dimensional GMV maps obtained from the pre-processing step were fed as input features to a 3D adaptation of the AlexNet architecture, which is based on convolutional neural networks (CNN) [18], [22]. The architecture as described in Figure 3 takes GMV maps of size 121 x 145 x 121 as the input, followed by five blocks of 3D CNN layers, with channel widths of 64, 128, 192, 192, and 128. Each block had batch normalization, rectifying linear unit (ReLU) function, and max-pooling layers applied to it. In the second part of the architecture, the features encoded by the CNN blocks were fed into a set of fully connected layers as shown in Figure 3. In the last fully connected layer, the number of output nodes was set to be the same as the number of classes (=2) for the classification task. The experiments were conducted using the PyTorch library in Python on NVIDIA Tesla V100 32GB GPUs, with an average training time of 2.5 hours per repetition.

We used the Adam optimizer with a batch size of 32 and an initial learning rate of 1e-3 for up to 200 epochs, followed by an early stopping condition with a patience level of 20 epochs to prevent overfitting. The aforementioned architecture was trained on the ADNI data with a Bayesian hyperparameter tuning on the batch size and learning rate. To get more robust measures, we utilized a repeated stratified k-fold cross-validation procedure (n,k=10) to obtain repeated results by optimizing the model of training and validation sets and recording the performance on non-overlapping held-out test sets covering the whole dataset across the repetitions.

We computed whole-brain saliency maps for the full dataset by using gradient back-propagation on test subjects from each repetition. The saliency maps were aggregated together to be subsequently used as input for the constrained source-based salience (cSBS) analysis.

### C. Active Subspace Learning

For a given point $\mathbf{x} \in \mathbb{R}^m$ in the space of input features with $m$-dimensions, let $f : \mathbb{R}^m \to \mathbb{R}$ be a function mapping the input space to the space of the target variable(s). In the context of neuroimaging data, $x$ could represent structural features from the brain like GMV maps and the target variable $y$ could be a clinically observed cognitive or biological assessment. In such a scenario, the function $f$ can correspond to a classifier trained on the data. Active subspace learning [23], [24] for the given mapping $f$ involves an eigendecomposition of covariance of the gradients of $f$. The covariance matrix $\mathbf{C}$ is defined as:

$$\mathbf{C} = \mathbb{E}\left[(\nabla_x f)(\nabla_x f)^T\right] \tag{1}$$

$$\hat{\mathbf{C}} = \frac{1}{n}\sum_{i=1}^{n}(\nabla f(\mathbf{x}_i))(\nabla f(\mathbf{x}_i))^T \tag{2}$$

$\mathbf{C}$ can also be estimated as $\hat{\mathbf{C}}$ using the data samples of size $n$. In the current context, $f$ is the underlying function representing the trained 3D-CNN architecture in Figure 3.

Subsequently, the eigendecomposition step computes the eigenvectors of $C$ with significantly large eigenvalues (Equation 4) to represent the set of active subspaces. The saliency features $\mathbf{G} = \nabla f(\mathbf{X})$ can be projected onto the active subspaces to obtain transformed saliency features $\hat{\mathbf{G}}$ (Equation 5).

$$\mathbf{C} = \mathbf{W}\mathbf{\Lambda}\mathbf{W}^T \tag{3}$$

$$\mathbf{\Lambda} = \begin{bmatrix} \mathbf{\Lambda_A} & \\ & \mathbf{\Lambda}_I \end{bmatrix}, \; \mathbf{W} = [\mathbf{W_A} \; \mathbf{W}_I], \tag{4}$$

$$\text{such that } \mathbf{\Lambda}_I \approx \mathbf{0}, \text{and} \;\; \lambda_i \gg 0 \; \forall \lambda_i \in \mathbf{\Lambda}_A$$

$$\hat{\mathbf{G}} = (\mathbf{W_A})^T \mathbf{G} \quad\;, \text{ where } \mathbf{G} = \nabla f(\mathbf{X}) \tag{5}$$

### D. scICA for constrained source-based salience (cSBS)

It can be noted that the computing of active subspaces described in subsection II-C is equivalent to principal component analysis (PCA) in the saliency space of the data. This holds because the active subspaces are computed using the eigendecomposition of covariance in the saliency space, thus corresponding to mutually orthogonal directions in which the saliency has the highest variance. In the context of structural neuroimaging data, it amounts to an automated characterization of multi-faceted structural brain changes that drive the associations of brain structure with the clinically observed diagnostic variable.

In the case of voxel-level neuroimaging features, independent component analysis (ICA) applied to the PCA output has been shown to be more stable for computing meaningful sources from large neuroimaging data, because it also involves higher order statistics than just non-correlation maximized by PCA [8]. As a next step, we use ICA on the active subspace projections defined as $\hat{\mathbf{G}}$ in Equation 5, as follows:

$$\hat{\mathbf{G}} = \mathbf{A}\mathbf{S} \tag{6}$$

$$\hat{\mathbf{S}} = \mathbf{W_A}\mathbf{S}^T \tag{7}$$

**A** represents the ICA loading matrix and **S** corresponds to the ICA source components defined as a weighted linear combination of active subspaces in $\mathbf{W_A}$. Thereafter, full-brain independent source-based salience maps are computed via back-reconstruction Equation 7, which signify multiple independent salient brain components that capture the most discriminative structural brain changes for a given clinically observed categorical variable (AD diagnosis).

To make the aforementioned ICA analysis more robust and interpretable, we employed a spatially-constrained ICA (scICA) approach available in the GIFT toolbox (https://trendscenter.org/software/gift/) [8]. This approach performs ICA on a given set of full-brain maps to estimate source-based components and loadings, but with spatial constraints such that the component estimation process is guided by available brain network templates [9], [11]. We used the NeuroMark template [9] available in the GIFT toolbox, for computing the spatially constrained salient components. Essentially, the scICA approach uses pre-existing spatial brain templates as reference during the ICA procedure to produce components that are meaningful and interpretable in terms of brain regions.

Thus, the overall process of constrained source-based salience (cSBS) as shown in Figure 3 analysis yields cSBS components and subject-specific cSBS loadings, where the components can be interpreted to signify the salient sources for the diagnostic classification and the loadings encode the extent of their salience towards the classification process.

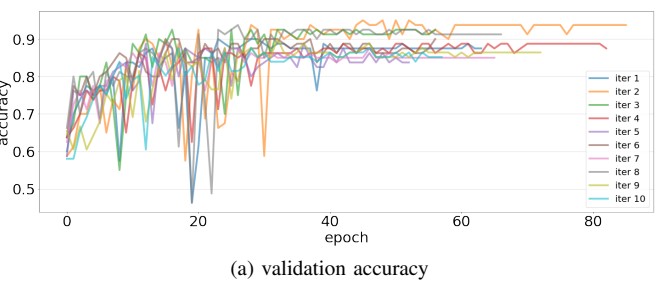

(a) validation accuracy

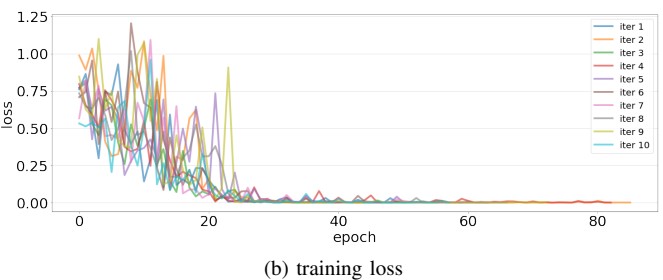

(b) training loss

Fig. 2: Training learning curves with validation accuracy and training loss (cross-entropy) are plotted for the CNN-based deep learning architecture trained in the first step (Figure 3) on the AD-classification task. The training was done for up to 200 epochs with an early stopping criteria (patience = 20 epochs). The plots are shown for each repetition of the analysis done with a 10-fold mutually exclusive train-test split, each with a 10-fold internal validation.

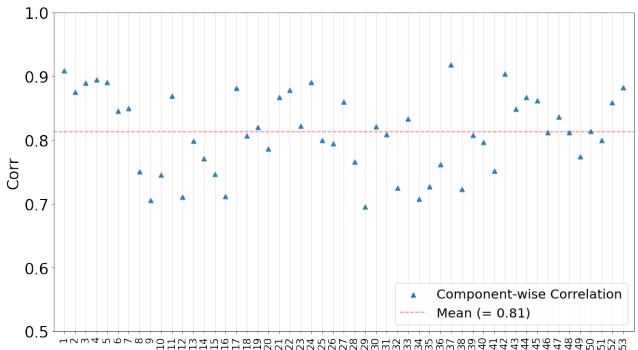

Fig. 3: Correlation of constrained source-based salience (cSBS) components with the 53 NeuroMark components [9]. Neuromark was used as a template to guide the spatially constrained ICA procedure (step 4 in Figure 3) for computing 53 cSBS components such that they correspond to meaningful brain areas. It can be noticed that the cSBS components computed with spatial constraints are very similar to the corresponding Neuromark components (mean correlation = 0.81, std=0.06).

## III. RESULTS

### A. Model Training

With hyperparameter tuning for batch size (bs) and learning rate (lr), the model described in subsection II-B was trained with hyperparameters (bs=8, lr=2.35e-5) with external 10-fold cross-validation for 10 train-test repetitions. Within each repetition, $10\%$ samples were used for internal validation during training. Figure 2 shows the learning curves for validation cross-entropy loss and accuracy scores for 10 repetitions of the training procedure. The learning curves for the training loss and validation accuracy during the training process stabilize to an optimal range of values for different model configurations that are encountered after about 50 epochs of training, indicating that the model performance remains stable under different parametric configurations during the training. The final test accuracy (mean = $90.65\%$, std= $4.1\%$) was almost the same as previous studies involving the use of CNN-based architectures for AD classification from sMRI-based full-brain features [20]. It is worth mentioning that the focus of this study is primarily to compute meaningful cSBS maps, given that the model performance is comparable to existing baseline values.

### B. Computing cSBS Components

The saliency maps computed via gradient back-propagation on the trained models were used as input for the constrained source-based salience (cSBS) analysis described in subsection II-D. We used the spatially constrained ICA (scICA) decomposition function available in GIFT toolbox (https://trendscenter.org/software/gift/) [8] which is implemented in such a manner that it incorporates both the PCA step for active subspaces (subsection II-C) and the subsequent scICA step for cSBS analysis (subsection II-D), returning the back-

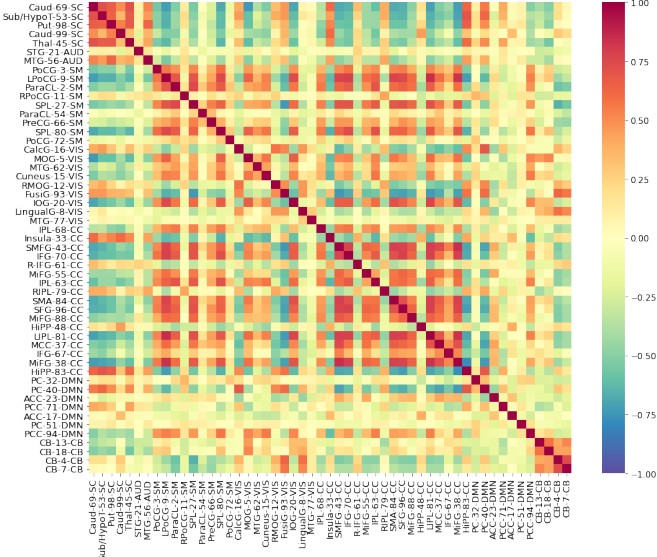

Fig. 4: Functional Network Connectivity (FNC) matrix for the cSBS loadings, computed as the pair-wise Pearson correlation between the loadings corresponding to each pair of cSBS components. The components are marked (eg. Insual-33-CC) with the corresponding brain area, serial number, and functional sub-domain in the Neuromark template (refer [9] for a detailed table). The functional sub-domains that the components are divided into are the default mode network (DMN), visual areas (VIS), auditory areas (AUD), cerebellar areas (CB), cognitive control (CC), sensorimotor (SM) and sub-cortical (SC) areas.

reconstructed cSBS brain components $\hat{\mathbf{S}}$ (Equation 7), and the corresponding subject-specific cSBS loadings $\mathbf{A}$ (Equation 6).

The Neuromark template was used as the guidance for the spatial constraints in the scICA step with 53 intrinsically connected networks (ICNs), corresponding to different brain areas, further divided into 7 sub-domains based on their functions: the default mode network (DMN), visual areas (VIS), auditory areas (AUD), cerebellar areas (CB), cognitive control (CC), sensorimotor (SM) and sub-cortical (SC) areas. More details about the Neuromark template can be found in [9]. While alternate templates could have been utilized in the context of such frameworks, we chose to use the Neuromark template because it is also based on a robust group-ICA based decomposition framework to compute the brain components [9]. Moreover, the 53 components have been manually verified in the case of Neuromark template by clinical experts to correspond to meaningful brain regions. While the effect of various available templates can be explored in detail as a separate study, previous studies involving the use of subspace decomposition without deep learning [13] or even DL-based saliency analysis without decomposition [25] have used ROI-based atlases and found similar results in terms of the involved brain areas.

PCA and scICA steps using Neuromark as the reference template for spatial constraints were performed with a model

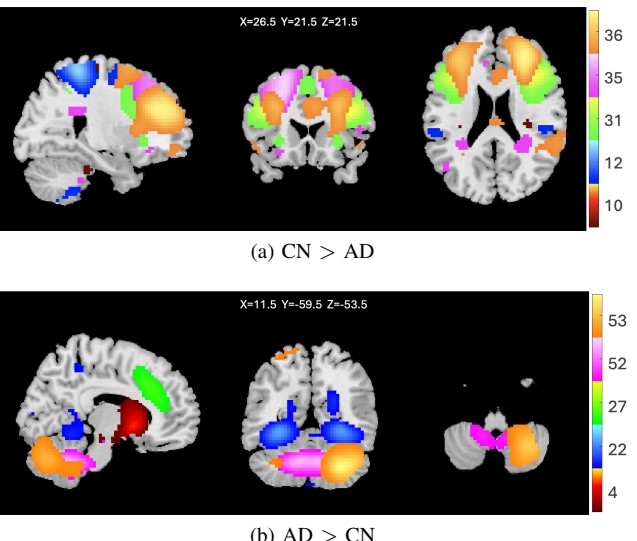

(a) CN > AD

(b) AD > CN

Fig. 5: Top 5 cSBS components with most significant group differences (CN vs AD) in discriminative saliency towards AD diagnosis. The final back-reconstructed components computed using the scICA step in the cSBD analysis (Figure 3) are visualized in the standard MNI voxel space after standardization and thresholding the z-values ($|z| > 3$). These components represent the brain areas in which structural brain changes are the most distinctive toward (a) CN more than AD group, and (b) AD more than CN group.

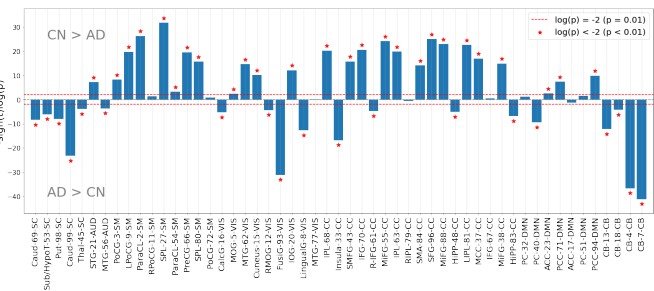

Fig. 6: Group comparison of cSBS loadings. The plot shows $-sign(t)\log(p)$ from a two-sample t-test performed between CN and AD groups. The cSBS procedure can extract components such that most of them show significant group differences in their corresponding loadings ($p < 0.01$, i.e. $|-sign(t)\log(p)| > 2$).

order of 100 and 53 respectively, resulting in 53 cSBS components ($\hat{\mathbf{S}}$) and corresponding subject-specific cSBS loadings ($\mathbf{A}$). The model order of 53 was selected to match the number of components in the Neuromark reference template consisting of robust brain components that have also been manually verified to correspond to biologically meaningful brain areas. The resultant components were similar to the corresponding Neuromark ICNs, each with a high correlation (mean=0.81, std=0.06), as shown in Figure 5.

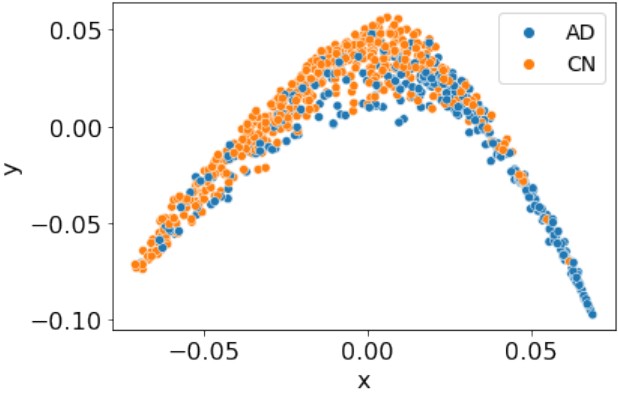

(a) LLE visualization of cSBS loadings

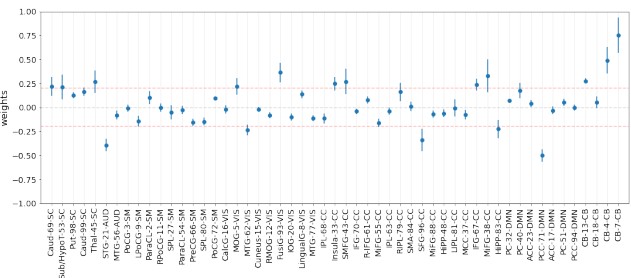

(b) Prediction weights for CN vs AD classification

Fig. 7: (a) Local linear embedding (LLE) to create a 2-D visualization of cSBS loadings from the most discriminative components based on significant group differences $| - sign(t) \log(p)| > 20$. (b) Prediction weights for CN vs AD classification using logistic regression on 53-dimensional subject-specific cSBS loadings.

### C. cSBS Loadings and their Biological Relevance

Figure 6 shows side-by-side boxplots for the 53-dimensional cSBS loadings for CN and AD groups, along with $-sign(t) \log(p)$ values for two-sample t-test on CN vs AD loadings for each of 53 cSBS components. It can be noted that the cSBS procedure is able to extract meaningful brain components and loadings with significant group differences ($p < 0.01$, i.e. $| - sign(t) \log(p)| > 2$) for most of the components. This indicates the capability of the cSBS framework to automatically uncover discriminative brain areas underlying salient structural differences in CN vs AD subjects.

Figure 4 shows the functional network connectivity computed as pair-wise Pearson correlation between cSBS loadings from each of the 53 components. It can be noted that the loadings, that capture the strength of the salient contribution of cSBS brain components, are inherently organized into multiple functional brain networks based on the connectivity patterns. This is indicated by the high intra-network correlation among the components within subcortical (SC), visual (VIS), cognitive control (CC), sensorimotor (SM), and Cerebellar (CB). All of these functional sub-domains of the brain are known to be affected in AD [26]. Components corresponding

to brain areas from these domains also feature among the set of components with stronger group differences in cSBS loadings ($| - sign(t) \log(p)| > 10$ in Figure 6). Another notable observation is the presence of strong inter-subdomain connectivity between CC-VIS, CC-SM, VIS-SM components (Figure 4). FNC interactions between these sub-domains have also been reported in AD-related functional MRI studies [27], [28]. Thus, our model is also able to capture the salient inter-subdomain structural patterns of collective change which are similar to known functional network connectivity changes in AD.

Figure 5 shows brain maps for the top 5 cSBS components with the strongest group differences based on the $-sign(t) \log(p)$ values. It is interesting to note that the top cSBS brain components with stronger loadings in AD group than CN group (AD > CN) are mainly from CB, VIS, and SC sub-domains, while the ones with stronger loadings in CN group than AD group (CN > AD) are from SM and CC domains. This could be due to a known mix of atrophy and inflammation in various brain areas in Alzheimer's disease, which leads to an increase and decrease in the GMV in these areas, respectively [29], [30].

Additionally, we also studied the predictive power of the cSBS loadings by using them for CN vs AD classification. We used logistic regression for this purpose, with a 10-fold external as well as internal cross-validation procedure with grid search. The resultant mean test accuracy was $82.25\%$ (with std = $2.6\%$), indicating the retainment of the predictive power of cSBS loadings. Figure 7 shows the locally linear embeddings (LLE) visualization of the top discriminative features among the sSBS loadings along with the component-wise weights of the learned classifier. It can be noted that the components with higher weights also correspond to the ones with the most significant group differences in Figure 6.

## IV. CONCLUSION

By introducing spatially constrained source separation in the saliency space, our work presents a methodological framework to summarize saliency information from deep learning models into interpretable brain component maps and their contributive strength towards driving the diagnostic discrimination. Upon training the CNN-based architecture for AD classification to achieve desirable performance, we show that the subsequent active subspace learning (ASL) and the spatially constrained ICA (scICA) steps yield meaningful brain components and corresponding loadings that represent the AD-related changes in structural gray matter volume features of the brain. Additionally, despite being computed from structural saliency maps, the cSBS loadings encode important AD-related connectivity patterns between and within functional sub-domains of the brain. Thus, the cSBS procedure utilizes ASL and scICA methods to explain trained deep models on MRI data in terms of decompositions that can be visualized in terms of existing brain templates, and encode important collective brain network changes into source-based salience maps while retaining predictive performance. Toward biomarker detection for neurolog-

ical disorders like AD, it is of utmost importance to develop such frameworks that can employ deep learning models for neuroimaging data toward a more robust understanding and visualization of the salient structural brain changes associated with the disorder. Such frameworks should be well-nuanced to summarize the multi-faceted changes involved in distinguishing neurological disorders but should also be robust enough to retain predictive association between neuroimaging features and diagnostic variables. Our framework is a step in the same direction.

In the near future, we plan to apply this framework to datasets from more neurological disorders and extend it to incorporate longitudinal datasets. This can potentially enable the tracking of multiple brain subsystems associated with longitudinal changes involved in the manifestation and progression of brain disorders. While the scope of this work was limited to the use of CNN-based deep learning architectures followed by a spatially-constrained ICA step in the saliency space, the use of other predictive deep learning architectures, saliency computation methods as well as decomposition techniques can also be explored as part of a larger comparative study in the future. It is also possible to develop extended versions of this framework for multimodal data to include functional neuroimaging features and genetic data.

## ACKNOWLEDGEMENTS

The research reported in this work was supported by the National Institutes of Health (NIH) under grant number NIH-R01AG073949.

Data used in preparation of this article were obtained from the Alzheimer's Disease Neuroimaging Initiative (ADNI) database (adni.loni.usc.edu). As such, the investigators within the ADNI contributed to the design and implementation of ADNI and/or provided data but did not participate in the analysis or writing of this report. A complete listing of ADNI investigators can be found at: http://adni.loni.usc.edu/wp-content/uploads/how_to_apply/ADNI_Acknowledgement_List.pdf

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
