# OpenReview forum: "Constrained Source-based Salience: Network-based Visualization of Deep Learning Neuroimaging Models"
_IEEE.org/EMBS/BHI/2024/Conference — IEEE BHI'24_

### Official Review · Reviewer_4W8B · 2024-08-06
**Review of "Constrained Source-based Salience: Network-based Visualization of Deep Learning Neuroimaging Models"**

**Overall Rating:** 6
**Confidence:** 2

**Other Quality Metrics:**

(a) Clarity of writing; Fair
 (b) Clinical Significance; Fair
(c) Methodological Novelty; Good
(d) Experiments and Results; Good

**Questions For The Authors:**

How does the cSBS framework compare with other state-of-the-art methods for analyzing neuroimaging data in terms of accuracy, interpretability, and clinical relevance?
Could you provide more details on the rationale behind choosing the specific components and parameters for the cSBS framework, such as the number of components used in the ICA step?
The paper mentions the use of existing brain network templates for spatial constraints. How sensitive is the cSBS framework to the choice of these templates, and could alternative templates lead to different results?

**Strengths:**

The central idea presented in this paper is intriguing and has significant clinical relevance. The proposed cSBS framework addresses a critical challenge in neuroimaging: the difficulty of interpreting complex deep learning models at the level of brain networks. By combining deep learning with established neuroimaging analysis techniques like scICA, the authors propose a method that could potentially bridge the gap between powerful predictive models and the need for biological interpretability. The application of the method to Alzheimer’s disease data is particularly noteworthy, as it could lead to better understanding and visualization of disease-related brain changes, potentially aiding in the development of biomarkers for neurological disorders.

**Summary Of The Paper:**

The paper introduces a novel framework called Constrained Source-based Salience (cSBS) aimed at improving the interpretability of deep learning models applied to high-dimensional neuroimaging data. Specifically, the method is designed to identify and visualize independently salient brain networks associated with clinical diagnostic assessments. The authors demonstrate the application of this framework in the context of Alzheimer’s disease (AD) classification, using convolutional neural networks (CNNs) to analyze structural MRI data. The framework involves active subspace learning (ASL) and spatially constrained independent component analysis (scICA) to extract meaningful components that can be linked to existing brain templates, thereby improving the interpretability of deep learning results in the field of neuroimaging.

**Weaknesses:**

While the central idea of the paper is compelling, the presentation of the content is confusing and difficult to follow, which affects the overall clarity of the work. The methodological details are dense and may overwhelm readers who are not deeply familiar with the specific techniques used. Additionally, the paper lacks a comparative analysis with other similar approaches, making it difficult to assess the novelty and importance of the contribution. The absence of discussion on how cSBS compares to existing methods in terms of performance, interpretability, and clinical applicability limits the paper’s impact. Moreover, the paper could benefit from a more structured discussion on potential limitations and the scope of generalizability of the proposed framework.

In addition, given the dataset used, I assume that the selection of the disease is more related to the availability of data rather than at the will of making improvements in the clinical analysis of AD. This reads mostly as a  machine learning exercise rather than biomedical research with clinical relevance.

---

### Official Review · Reviewer_UajQ · 2024-08-08
**Enhanced Interpretability in Deep Learning for Alzheimer's Diagnosis Using Constrained Source-Based Salience Framework**

**Overall Rating:** 6
**Confidence:** 3

**Other Quality Metrics:**

1. Clarity of Writing: Good
2. Clinical Significance: Good
3. Methodological Novelty: Fair
4. Experiments and Results: Good

**Questions For The Authors:**

1. While the framework was validated on an AD dataset, its applicability to other neurological disorders or different types of neuroimaging data remains to be demonstrated.
2. What are the specific computational requirements for implementing the cSBS framework, and are there any strategies to optimize its efficiency?

**Strengths:**

1. The integration of ASL and scICA in the saliency space of CNNs represents a significant methodological advancement, offering enhanced interpretability of deep learning models in neuroimaging.
2. The approach was validated using repeated analyses on a large dataset, ensuring the reliability and consistency of the findings.
3. The identified brain network components are consistent with known AD-related brain areas, underscoring the biological relevance and potential clinical utility of the framework.

**Summary Of The Paper:**

The paper introduces a novel constrained source-based salience (cSBS) framework designed to enhance the interpretability of deep learning models applied to high-dimensional MRI data for neurological disorder diagnosis, specifically Alzheimer's disease (AD). The framework integrates active subspace learning (ASL) and spatially constrained independent component analysis (scICA) within the saliency space of convolutional neural networks (CNNs). This approach allows for the automated identification and visualization of multiple salient brain networks, which are consistent and relevant to AD-related brain areas. The method was validated through analyses of the ADNI dataset, demonstrating the framework's capability to detect meaningful and interpretable brain network components.

**Weaknesses:**

1. The performance of the framework might be sensitive to the choice of parameters in ASL and scICA, and it is unclear how robust the framework is to variations in these parameters.
2. The framework's complexity may pose challenges for replication and implementation by other researchers, particularly those without a deep understanding of deep learning and neuroimaging techniques.

---

### Official Review · Reviewer_n3qa · 2024-08-13
**Constrained Source-based Salience: Network-based Visualization of Deep Learning Neuroimaging Models**

**Overall Rating:** 7
**Confidence:** 4

**Other Quality Metrics:**

a. Good
b. Excellent
c. Good
d. Excellent

**Questions For The Authors:**

1. Would this framework generalize to other neurological conditions?

2. Can you clarify if the salient brain networks identified offer insights into the underlying mechanisms of Alzheimer’s disease or if they primarily serve predictive purposes?

3. What are the computational requirements for the framework? For instance, what is the training time?

4.  What are the next steps in this research? Are there plans to validate these findings in clinical settings or incorporate longitudinal data to track disease progression?

**Strengths:**

1. The application to Alzheimer's disease, a condition with significant clinical and social impact, adds substantial value to the study. The ability to map and interpret brain changes directly associated with AD could improve diagnostic processes and potentially lead to better patient outcomes.

2. The use of a well-known dataset (ADNI) and rigorous preprocessing makes the experiments credible.

3. The paper provides extensive validation of the cSBS components through biological relevance and statistical tests, supporting their significance in the context of AD. The repeated training scenarios and cross-validation contribute to the robustness of the findings.

**Summary Of The Paper:**

This paper presents a novel framework: constrained source-based salience (cSBS), which combines deep learning with neuroimaging to identify and visualize salient brain networks linked to clinical assessments, particularly for Alzheimer's disease (AD). The authors utilize a combination of convolutional neural networks (CNNs) for classification tasks and subsequent subspace learning techniques—specifically active subspace learning (ASL) and spatially constrained independent component analysis (scICA) to extract meaningful brain components from MRI data. They verify these components against established brain network templates, aiming to improve interpretability in clinical diagnostics and biomarker identification.

**Weaknesses:**

1. The methods are complex and might be challenging for readers not familiar with both deep learning and neuroimaging. Simplifying the explanations or providing more intuitive visualizations could improve the paper.

2. While the framework helps in interpreting the CNN's outputs, there is a subtle distinction between interpretability and true explanation. The authors could discuss whether the identified components offer insights into the disease’s pathophysiology or merely correlate with clinical symptoms.

---

### Decision · Program_Chairs · 2024-09-23

Accept